# Economics of physical activity in low-income and middle- income countries: a systematic review

Priyanga Diloshini Ranasinghe [1,2] Subhash Pokhrel,[1] Nana Kwame Anokye[1]

¹Health Economics Research Group, Division of Global Public Health, Department of Health Sciences, Brunel University London, UB8 3PH, UK
²Ministry of Health, Nutrition, Indegenous Medicine, Colombo 10, Sri Lanka

**Correspondence to**
Dr Nana Kwame Anokye, Health Economics Research Group, Division of Global Public Health, Department of Health Sciences, Brunel University London, UB8 3PH, UK;
Nana.Anokye@brunel.ac.uk

## ABSTRACT

**Background** Promotion of physical activity (PA) among populations is a global health investment. However, evidence on economic aspects of PA is sparse and scattered in low-income and middle-income countries (LMICs).

**Objective** The objective of this study was to summarise the available evidence on economics of PA in LMICs, identify potential target variables for policy and report gaps in the existing economic evidence alongside research recommendations.

**Data sources** A systematic review of the electronic databases (Scopus, Web of Science and SPORTDiscus) and grey literature.

**Study eligibility criteria** Cost-of-illness studies, economic evaluations, interventions and descriptive studies on economic factors associated with PA using preset eligibility criteria.

**Study appraisal and synthesis of methods** Screening, study selection and quality appraisal based on standard checklists performed by two reviewers with consensus of a third reviewer. Descriptive synthesis of data was performed.

**Results** The majority of the studies were from upper-middle-income countries (n=16, 88.8%) and mainly from Brazil (n=9, 50%). Only one economic evaluation study was found. The focus of the reviewed literature spanned the economic burden of physical inactivity (n=4, 22%), relationship between PA and costs (n=6, 46%) and socioeconomic determinants of PA (n=7, 39%). The findings showed a considerable economic burden due to insufficient PA, with LMICs accounting for 75% of disability-adjusted life years (DALYs) globally due to insufficient PA. Socioeconomic correlates of PA were identified, and inverse relationship of PA with the cost of chronic diseases was established. Regular PA along with drug treatment as a treatment scheme for chronic diseases showed advantages with a cost–utility ratio of US$3.21/quality-adjusted life year (QALY) compared with the drug treatment-only group (US$3.92/QALY) by the only economic evaluation conducted in the LMIC, Brazil.

**Limitations** Meta-analysis was not performed due to heterogeneity of the studies.

**Conclusions and recommendations** Economic evaluation studies for PA promotion interventions/strategies and local research from low-income countries are grossly inadequate. Setting economic research agenda in LMICs ought to be prioritised in those areas.

**PROSPERO registration number** CRD42018099856.

### Strengths and limitations of this study

► This is the first synthesis of literature on all types of economic analyses of physical activity in LMICs.
► The study uses a comprehensive search strategy that covers literature from academic databases and websites of international organisations such as WHO.
► This review focuses on studies written in English language and could miss out on relevant literature published in other languages.
► Meta-analysis was not possible due to wide variation in focus and methods of included studies.

## INTRODUCTION

Regular physical activity (PA) is proven to have multiple health benefits, including preventing and treating non-communicable diseases (NCDs), as well as improving the mental health and quality of life.[1] PA is defined by WHO as 'any bodily movement produced by skeletal muscles that requires energy expenditure—including activities undertaken while working, playing, carrying out household chores, travelling, and engaging in recreational pursuits'.[2] WHO recommends at least 150 min of moderate-intensity PA or 75 min of vigorous-intensity PA throughout the week for adults/elderly and 60 min of moderate-intensity to vigorous-intensity PA daily for children aged 5–17 years.[2] Insufficient physical activity (IPA) or physical inactivity is defined as the absence or insufficient level of PA required to meet the current PA recommendations.[1]

Of all the lifestyle behaviours, PA has one of the largest positive impact on the risk of all-cause mortality (33% reduction).[3] IPA accounts for 3.2 million deaths annually (including 9% premature deaths), and in 2010, 69.3 million disability-adjusted life years (DALYs) were attributed to physical inactivity worldwide.[4] NCDs are a major global health challenge, and IPA is estimated to cause a large share of the burden of main NCDs: 6%

(3.2%–7.8%) of chronic heart disease burden, 7% (3.9%–9.6%) of type 2 diabetes burden, 10% (5.6%–14.1%) of breast cancer burden and 10% (5.7%–13.8%) of colon cancer burden.[5] Mortality attributed to NCDs is projected to increase from 38 million in 2012 to 52 million by 2030.[3] WHO reports that NCDs are disproportionately rising in low/middle-income countries (LMICs), and in 2016 over three-quarters of NCD deaths (31.5 million) occurred in LMICs, of which about 46% of deaths occurred before the age of 70 years.[6]

The prevalence of IPA among adults in 2016 was 27.5% (25·0%–32.2%) globally, 16·2% (14·2%–17·9%) in low-income countries (LICs) and 26·0% (22·6%–31·8%) in middle-income countries (MICs).[7] The rate was even higher among schoolgoing adolescents in LICs (84·9%) and MICs (81%).[8] The prevalence of IPA has been stagnant over the last 15 years in these countries.[7 8] To achieve the WHO target of 15% relative reduction in the prevalence of IPA by 2030 in these countries, urgent PA promotion is required. To date, progress of implementation has been plagued by lack of awareness and investment.[1] This is partially attributable to the paucity of evidence base on the economic gains accruable from promoting PA in LMIC settings.[9 10] There is an unmet demand for an evidence base on economics of PA in LMICs, which can be used as an advocacy tool to prioritise PA promotion. This is critical especially in LMICs, given the unlimited competing health needs and developing economies. Although there are few economic studies on PA, the evidence base is scattered. Systematic review is a high-quality source to inform formulation of policies and setting of budget priorities and to build a coherent research base,[9] but such studies are lacking in economics of PA. A few exceptions include a comprehensive review of studies estimating the economic burden of PA worldwide.[11] This study has a broader scope in terms of the economic evidence (all types of economic analyses) but with limited geographic settings. In line with the standard definition of economics,[12] we categorise economic research on PA into examining socioeconomic determinants of PA; impacts of physical (in)activity on economic outcomes such as costs, quality of life and labour market outcomes; and economic evaluation studies on PA interventions.[9 10] This broad perspective is to facilitate a broader mapping of the multiple facets of the evidence base in LMICs. A specific review of LMICs is required, considering the differing dynamics of the health systems[13] and the urgent need to better understand the research base and map gaps in knowledge given the growing burden of NCDs.[2] Health research in LMICs is warranted to overcome global health challenges.[14]

Thus, the aim of this review was to (1) summarise the available evidence on economics of PA in LMICs, (2) describe the focus and methods underpinning research on economics of PA in LMICs, (3) identify potential target variables and cost-effective interventions in LMICs for policy formation regarding PA promotion and (4) identify and report gaps in research on economics of PA

in LMICs and provide recommendations for economic research agenda in LMICs.

## METHODS
Study protocol with detailed methods of this review was published prior to the commencement of this review[15] and is shown in online supplemental file 1.

### Search strategy
Systematic literature search was conducted to identify studies on economics of PA in LMICs using the following databases: Scopus (covers 100% of MEDLINE coverage, 100% of EMBASE coverage, 100% of Compendex coverage[16]), Web of Science and SPORTDiscus. Websites of WHO, National Institute for Health and Care Excellence (NICE) international, and World Bank and reference lists of included studies were searched for any relevant studies. The search period covered beginning of records to December 2017 in line with the study protocol. To identify any publications after the search end date (2017), a more recent search was conducted in August 2020 using Scopus, the biggest bibliographic database. The search strategy was informed by a scoping review that covered relevant reviews on economics studies of PA,[17–23] reviews on PA[24–26] and reviews on economic evaluations.[27–29] The final search strategy including search terms for each database is given in online supplemental file 2.

### Eligibility criteria and screening
All results were screened by titles to remove irrelevant studies and duplications independently by two reviewers (PDR, NKA). Disagreements were discussed with a third reviewer (SP). The remaining studies were further screened by titles and abstracts prior to full-text screening. Studies were included if the following criteria were met: (1) study setting: any setting of LMICs in accordance with the definition of LMICs by World Bank classification in 2017[13]; (2) population: any age group across life course; (3) interventions: any PA intervention; (4) comparator: normal routine, no intervention (intervention and comparator applicable only for intervention studies); (5) outcomes: (a) cost-effectiveness ratio, quality-adjusted life year (QALY) and incremental cost-effectiveness ratio assessed as the outcome of PA intervention and (b) cost of physical (in)activity in terms of healthcare cost and/or productivity loss and/or total cost of physical inactivity; (6) study design: observational studies (cohort, case–control, cross-sectional), experimental studies including randomised control trials (RCTs), quasi-experimental studies including natural experiments and economic evaluation studies; and (7) studies reported in English language.

Case reports, case series, letters to the editor, editorials, reviews, qualitative studies, unpublished thesis, conference abstracts and any unobtainable texts were excluded. Studies that did not report cost, economic evaluations and

economic correlates of PA, separately from other lifestyle risk factors, were excluded from the review by consensus of all three reviewers. The list of excluded studies with reasons is given in online supplemental file 3.

## Data extraction

Data on the following main items—general information, characteristics of the study, characteristics of population/intervention/comparator/outcome/study design (PICOS), data sources, analysis, results, conclusions and limitations—were extracted using a word template. Data extraction form was adapted from those of relevant reviews.[21 30 31] The full data extraction form is shown in online supplemental file 4.

## Risk of bias and quality appraisal

The checklists by Larg and Moss[32] and NICE[33] were used to assess the risk of bias of cost-of-illness studies and studies on correlations or associations, respectively. The checklists by Drummond and Jefferson[34] and Philips *et al*[35] were used for economic evaluation studies. The latter was used more specifically for model-based economic evaluations.

Overall quality grading was given for both internal validity and external validity using NICE recommendations.[33] The internal validity of studies on correlations was assessed based on sections 2–4 and external validity based on section 1 of the NICE checklist.[33] For cost-of-illness studies, external validity was assessed by Q2b(III) and internal validity by the remaining questions in the Larg and Moss checklist.[32]

Studies meeting all or most criteria (>75%) were scored '++', those meeting some criteria (50%–75%) were scored '+' and those meeting few criteria (<50%) were scored '−'. The cut-offs were set with full agreement of all three reviewers prior to commencement of the study.

Data from included studies were extracted by one reviewer (PDR). A second reviewer (NKA) independently extracted data from 50% (9/18) of the included papers, randomly selected. Disagreements were discussed with a third reviewer (SP). The same pattern was repeated for quality appraisal.

## Data synthesis

Descriptive synthesis of data was performed to describe the methods, operationalisation of methods, quality, major limitations, outcome, suggestions and recommendations for future research. Data from research evidence on economics of PA in LMICs were used to prepare summary tables. Identification of potential target variables for policy and gaps in research on economics of PA was gleaned from descriptive analysis of the available evidence.

## Patient and public involvement

The review was based on already available published research. The patients or the public were not directly involved in designing and conduct of the study. The results of the review will be disseminated through this scientific

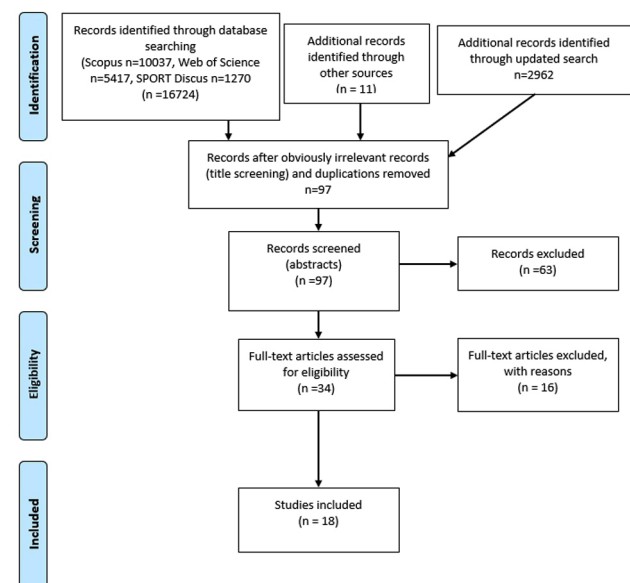

**Figure 1** Flow diagram of the article selection process. From Mohef *et al*.[58]

publication, conference presentations and social media networks available in public domain.

## RESULTS

### Search outcomes

The systematic literature search (in 2018) retrieved 16 724 records from the initial database search, 11 records from other sources and 2962 records from updated search (2020). After removing duplicates and irrelevant records by screening for titles, 97 articles were selected for abstract screening. Thirty-four records were assessed for eligibility at full-text screening. Eighteen articles were selected for data extraction.[36–53] The flow diagram of this process is illustrated in figure 1.

### Overview of studies

The 18 studies originated from Brazil (n=9, 50.0%), China (n=3, 16.7%), Malaysia (n=2, 11.1%), South Africa and Iran (n=1, 5.6% from each). There were two global analysis (n=2, 11.1%) (figure 2). The majority of the studies were conducted in upper-middle-income countries (n=16, 88.8%).[36 37 39–51 53] All studies were published after 2012. Online supplemental file 5 provides an overview of the aim of the study. Four studies focused on the economic burden of physical inactivity[38 43 47 51] (n=4, 22.2%), six studies assessed the association of physical (in)activity and direct or indirect cost[36 39 40 42 44 46] (n=6, 33.3%) and seven studies examined the association of physical (in)activity and economic factors[37 41 45 48–51] (n=7, 38.8%). Only one economic evaluation study was identified (n=1, 5.6%).[53]

### Methods of reviewed studies

The summary of studies by methods is given in table 1. The majority of the studies have used cross-sectional design (n=14, 77.8%).[36–44 47–49 51 52]

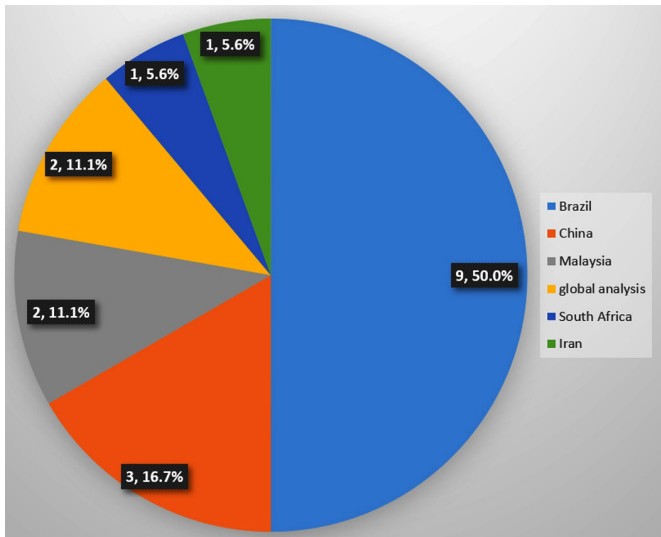

**Figure 2** Number of articles on economics of physical activity by country.

All four studies on economic burden of PA used secondary data to estimate the cost of inactivity, whereas the prevalence data on inactivity were obtained from population surveys or Global Burden of Disease study.

DALYs lost due to physical inactivity in LMICs were estimated by one study as a part of a global analysis.[38] Similar methodology was applied in the other three studies using population attributable fraction (based on prevalence of inactivity, relative risk (RR) values, cost of main NCDs related to physical inactivity).[38 43 51] Estimates for RR were obtained from developed countries due to unavailability

of data for LMICs. One study has assessed only the physical inactivity-related inpatient costs of chronic NCDs.[47]

The majority of studies examined the association of physical (in)activity and economic factors. The economic variables identified were direct health cost,[39 40 42 44 46] productivity lost[36] and socioeconomic indicators.[37 41 45 48 50 52] The method of analyses was regression-based particularly for the direct healthcare cost and productivity loss analyses[36 40 42 44 46] and bivariate for the rest.[37 41 45 48–50 52] A summary of included studies by PICOS is shown in table 2.

Table 3 shows that the method of operationalisation of PA varied across the studies. Only two studies used the WHO-recommended definition of sufficient PA—150 min of PA per week.[38 39] One study used more strict definition of sufficient PA activity—300 min during the last 7 days.[45] All PA data were based on self-reports, with the Baecke questionnaire identified as the most frequently used survey instrument.[36 40 42 44 53]

Cost was measured either as direct healthcare cost or as indirect cost. Direct healthcare cost was specified in practice as cost for medications, laboratory tests and medical consultations[39 40 42 44] and/or inpatient cost[39 46] and/or outpatient cost or prescription cost[38] (table 3). Indirect cost was operationalised as loss of productivity due to premature mortality,[38] absenteeism and disability retirement.[36]

### Quality appraisal of included studies

Table 4 summarises the quality appraisal of included studies, and online supplemental file 6 details the results

| **Table 1** | Summary of studies by focus and study design | |
|---|---|---|
| | **Number (n)** | **studies** |
| Focus | | |
| Economic burden of physical inactivity | 4 | Ding et al,[38] Zhang and Chaaban,[43] Bielemann et al,[47] Popkin et al[51] |
| Association of physical (in)activity and direct health cost | 5 | Mitsunaga et al,[39] Turi et al,[40] Codogno et al,[42] Turi et al,[44] Patel et al[46] |
| Association of physical (in)activity and productivity lost | 1 | Araujo et al[36] |
| Socioeconomic factors associated with physical activity participation and intensity | 7 | Cheah et al,[37 41] Silva,[45] Chen et al,[48] Sá et al,[49] Abdi,[50] Atkinson et al[52] |
| Economic evaluations related to physical activity | 1 | Queiroz et al[53] |
| Study design | | |
| Cross-sectional | 14 | Araujo et al,[36] Cheah et al,[37] Ding et al,[38] Mitsunaga et al,[39] Turi et al,[40] Cheah et al,[41] Codogno et al,[42] Zhang and Chaaban,[43] Turi et al,[44] Bielemann et al,[47] Chen et al,[48] Sá et al,[49] Popkin et al,[51] Atkinson et al[52] |
| Ecological/correlation study | 2 | Silva,[45] Abdi[50] |
| Longitudinal | 1 | Patel et al[46] |
| Intervention studies | 0 | |
| Economic evaluation studies | 1 | Queiroz et al[53] |

**Table 2** Summary of studies by PICOS

| Author year | Country/country category | Type of economic analysis | Population | Intervention | Comparator | Outcome | Study design |
|---|---|---|---|---|---|---|---|
| Araujo et al[36] | Brazil—UMC | Economic correlates of PA | ≥50-year-old adults | NA | NA | Strength of association between physical inactivity and productivity loss (OR, 95% CI) | Cross-sectional |
| Cheah et al[37] | Malaysia—UMC | Economic correlates of PA | General population | NA | NA | Strength of association of income and PA (coefficient of income) | Cross-sectional |
| Ding et al[38] | Global analysis—UMC/LMC/LIC | Economic burden | General population | NA | NA | Cost estimates, DALY | Cross-sectional |
| Mitsunaga et al[39] | Brazil—UMC | Association of PA programme with medical hospital cost | Employees of a private healthcare company | NA | NA | Correlation of programme attendance frequency with cost per capita. Association of programme adherence level with the average cost per capita/year | Cross-sectional (study design not specified in the original article. Based on the study pattern and type of analysis, we categorised as cross-sectional) |
| Turi et al[40] | Brazil—UMC | Association of PA and economic costs | ≥50-year-old adults | NA | NA | Association of walking and healthcare expenditure (OR, 95% CI) | Cross-sectional |
| Cheah et al[41] | Malaysia—UMC | Economic correlates of PA | Urban dwellers | NA | NA | Strength of association between participation decision, amount of PA with income, time, employment and insurance (coefficient) | Cross-sectional |
| Codogno et al[42] | Brazil—UMC | Association of PA with economic costs | ≥50-year-old adults | NA | NA | Strength of association between physical inactivity and healthcare expenditure (OR, 95% CI) | Cross-sectional |
| Zhang and Chaaban[43] | China—UMC | Economic burden | General population | NA | NA | Cost estimate of physical inactivity | Cross-sectional |
| Turi et al[44] | Brazil—UMC | Association of PA and economic cost | ≥50-year-old adults | NA | NA | Strength of association of PA and outpatient expenditure (OR, 95% CI) | Cross-sectional |
| Silva[45] | Brazil—UMC | Economic correlates of PA | Year 9 students public and private elementary schools | NA | NA | Strength of association of macroeconomic variables with PA (Pearson's r, 95% CI) | Correlation study |
| Patel et al[46] | South Africa—UMC | Association of PA with economic cost | Adults who are members of discovery health medical plan | Incentive-based health promotion programme | Not enrolled in the programme | Changes in level of participation in fitness-related activities, subsequent probability of hospital admissions and cost | Retrospective observational study. Not a intervention study |
| Bielemann et al[47] | Brazil— UMC | Economic burden | Above 40-year-old adults, Brazil | NA | NA | Inpatient cost estimates attributed to PA | Cross-sectional |
| Chen et al[48] | China—UMC | Economic correlates of PA | Permanent residents aged 30–65 years, Jiaxing, China | NA | NA | Strength of association of socioeconomic level and PA | Cross-sectional |
| Sá et al[49] | Brazil—UMC | Economic correlates of PA | ≥14-year-old general population, Brazil | NA | NA | Strength of association of socioeconomic status and active transport | Cross-sectional |

**Table 2** Continued

| Author year | Country/country category | Type of economic analysis | Population | Intervention | Comparator | Outcome | Study design |
|---|---|---|---|---|---|---|---|
| Abdi[50] | Iran—UMC | Economic correlates of PA | University students in Teheran universities | NA | NA | Strength of association of time, cost and attractiveness with LTPA | Correlational survey |
| Popkin et al[51] | China—UMC | Economic burden | General population | NA | NA | Cost estimate of PA | Cross-sectional (case study developed from a review) |
| Atkinson et al[52] | Global analysis—UMC/LMC/LIC | Economic correlates of PA | General population | NA | NA | Strength of association (OR) of PA and macroeconomic variables in 47 LMICs | Cross-sectional |
| Queiroz et al[53] | Brazil—UMC | Cost–utility analysis | General population | NA | NA | Cost–utility and incremental cost–utility of the treatments involved in PA compared with drug-only treatment group. QALY values and the mean cost of 12 months for each treatment scheme | Economic evaluation study |

DALY, disability-adjusted life year; LIC, low-income country; LMC, lower-middle-income country; LTPA, leisure time physical activity; PA, physical activity; PICOS, population/intervention/comparator/outcome/study design; QALY, quality-adjusted life year; UMC, upper-middle-income country.

of the quality appraisal by the Larg and Moss check-list,[38 43 47 51] the NICE checklist for studies on correlations and association[36 37 39–42 44–46 48–50 52] and the Drummond and Jefferson checklist for the economic evaluation study.[53] The majority of the studies fulfilled most of the criteria (n=11, 61.1%),[36–38 40–42 44 45 47 48 52] whereas five studies fulfilled some of the criteria (6/18, 33.3%)[43 47 49 50 53] for internal validity. Insufficient information on methods precluded quality appraisal for one study.[51] Similarly, external validity was good in all studies.

## Empirical results from reviewed studies
The empirical evidence from the reviewed papers is summarised below in the account of the main aims of the papers (table 5).

## Economic burden of physical inactivity in LMICs
### Direct cost
The direct healthcare cost in LMICs attributable to physical inactivity was found to be INT$10.3 billion in 2013, which is 19.1% of direct healthcare cost attributable to physical inactivity worldwide. Upper-middle-income countries contributed to 86% of this economic burden.[38] The estimated direct health cost in China attributed to physical inactivity in 2007 was US$3.5 billion,[43] whereas the cost of hospital admissions in Brazil in 2013 was estimated as US$0.73 billion by local studies.[47] The estimated direct healthcare cost attributed to physical inactivity was INT$3.1 billion for China and INT$1.6 billion for Brazil in 2013 by global analysis.[38] Inpatient cost and outpatient cost attributed to physical inactivity in 2000 was US$0.4 billion and US$0.9 billion, respectively, in China.[51] The

cost of chronic diseases attributed to physical inactivity in China accounted for 15.2% of the total cost of main NCDs in 2007, whereas 15% of inpatient costs from the Brazilian Health Care System in 2013 were related to insufficient PA.

## Indirect cost and DALYs
Physical inactivity-related deaths in LMIC settings contributed to productivity loss of INT$5.3 billion, covering 38.7% of productivity loss due to deaths related to physical inactivity worldwide.[38] LMICs bear 10.1 million DALYs attributed to physical inactivity, which is 75% DALYs worldwide.[38] The indirect cost of physical inactivity in China was estimated as US$3.3 billion in 2007.[43] The estimated indirect cost due to productivity loss by deaths related to physical inactivity in China was reported as INT$1.78 billion.

## Association of physical (in)activity and healthcare cost in LMICs
Walking (always) during leisure time was shown to be associated with lower healthcare expenditure in primary care (OR$_{adj}$=0.59, 95% CI=0.39 to 0.89).[40] Higher healthcare expenditure for medicine was associated with lower participation in PA at work (OR=1.58, 95% CI=1.06 to 2.35) and sport (OR=1.57, 95% CI=1.12 to 2.18) in Brazil.[42] Furthermore, expenditure related to medicine (r=0.109, 95% CI=0.046 to 0.171) and overall expenditure (r=0.092, 95% CI=0.046 to 0.171) were significantly associated with physical inactivity in Brazil.[42]In addition, a significant association of expenditure on medication was with moderate PA showed by another Brazilian study

**Table 3** Methods of operationalisation of physical activity and cost in economic studies of physical activity

| | Number | studies |
|---|---|---|
| **Method of operationalisation of physical activity** | | |
| By WHO definition | 1 | Ding et al,[38] Mitsunaga et al[39] |
| Physical activity for >300 min during the last 7 days | 1 | Silva[45] |
| Based on IPAQ | 1 | Atkinson et al[52] |
| By Baecke questionnaire | 3 | Araujo et al,[36] Turi et al,[40] Codogno et al,[42] Turi et al,[44] Queiroz et al[53] |
| Self-reported vigorous or moderate physical activity during the last 7 days | 2 | Cheah et al,[37 41] |
| Based on the number of gym visits per annum | 1 | Patel et al[46] |
| Moderate or vigorous physical activity during leisure | 1 | Abdi[50] |
| Performing any physical activity during leisure | 1 | Bielemann et al[47] |
| Walking or cycling to work regardless of the duration | 1 | Sá et al[49] |
| Not reported | 1 | Zhang and Chaaban[43] |
| **Operationalisation of cost in economic studies of physical activity** | | |
| **Method of operationalisation of direct cost** | | |
| Cost of disease using population attributable fraction related to physical inactivity | 3 | Ding et al,[38] Zhang and Chaaban[43] |
| Inpatient cost | 1 | Bielemann et al[47] |
| Total cost of consultations, diagnostic tests, hospital admissions, therapeutic procedures, emergency care paid by insurance | 1 | Mitsunaga et al[39] |
| Total cost of medication dispensed/laboratory tests/medical consultations | 3 | Turi et al,[40] Codogno et al,[42] Turi et al[44] |
| **Method of operationalisation of indirect cost** | | |
| Loss of productivity due to premature mortality | 1 | Ding et al[38] |
| Loss of productivity due to absenteeism and disability retirement | 1 | Araujo et al[36] |

IPAQ, International Physical Activity Questionnaire.

(OR=0.56, 95% CI=0.38 to 0.81).[44] A retrospective longitudinal study in South Africa showed that the odds of hospital admissions were 13% lower for two additional gym visits per week among participants of incentive-based fitness-related activity programme.[46] Members who were not on a health promotion programme had a significantly higher probability of hospital admissions (p<0.01) and higher claims than participants who changed their PA status to 'active to no change' group or 'active to more active group' in South Africa.[46] However, a Brazilian study showed inconclusive results for association of adherence to an incentive-based PA programme and direct healthcare cost among employees of a private healthcare company.[39]

Physical inactivity was found to be correlated with productivity losses. A Brazilian study showed that PA was associated with productivity loss due to absenteeism (OR$_{adj}$=0.29, 95% CI=0.09 to 0.88).[36]

### Economic factors associated with physical (in)activity in LMICs

A number of economic factors were shown to be potential target variables for PA promotion policies.

**Table 4** Quality appraisal of included studies

| | | Number of studies | Studies |
|---|---|---|---|
| By NICE checklist for quantitative studies on correlations/associations (n=13) | | | |
| Internal validity (sections 2–4) | ++* | 10 | Araujo et al,[36] Cheah et al,[37] Turi et al,[40] Cheah et al,[41] Codogno et al,[42] Turi et al,[44] Silva,[45] Patel et al,[47] Chen et al,[48] Atkinson et al[52] |
| | +† | 3 | Mitsunaga et al,[39] Sá et al,[49] Abdi[50] |
| | −‡ | 0 | |
| External validity (section 1) | ++ | 9 | Turi et al,[40] Mitsunago et al,[39] Cheah et al,[41] Codogno et al,[42] Turi et al,[44] Silva,[45] Patel et al,[46] Chen et al,[48] Atkinson et al[52] |
| | + | 3 | Araujo et al,[36] Cheah et al,[37] Abdi[50] |
| | − | 0 | |
| Checklist by Larg and Moss for cost-of-illness studies (n=4§) | | | |
| Internal validity | ++ | 1 | Ding et al[38] |
| | + | 2 | Zhang and Chaaban,[43] Bielemann et al[47] |
| | − | 0 | |
| External validity (Q2b(III)) | ++ | 0 | |
| | + | 1 | Ding et al,[38] Zhang and Chaaban,[43] Bielemann et al[47] |
| | − | 0 | |
| Checklist for economic evaluation studies by Drummond and Jefferson[34] (n=1) | | | |
| Validity | + | 1 | Queiroz et al[53] |

*(++) All or most checklist criteria (>75%) fulfilled.
†Some checklist criteria (50%–75%) fulfilled.
‡(−) None or few checklist criteria (<50%) fulfilled
§The work by Chen et al[48] is a case study based on a review; thus, information on methods is not adequate to report on the quality.
NICE, National Institute for Health and Care Excellence.

### Socioeconomic status

Two Malaysian studies revealed that individuals with higher levels of income are less likely to participate in PA.[37 41] This was corroborated by a Brazilian study using data from the Brazil's National Household Sample Survey 2008 to analyse income quintiles by active commuting. The study revealed that in urban population the lowest income quintile had higher active transport (walking and cycling) level than the wealthiest population (2–5 times higher)[49] (p<0.001). Similarly, in China income levels were associated with lower PA.[48]

The cross-sectional study carried out in China reported that people from upper socioeconomic status (≥12 index scores—score was based on educational attainment, occupation and income per capita) use active transport more compared with lower social classes (p<0.0001).[48] However, this study further revealed that people in upper socioeconomic classes had less overall intensity of PA in metabolic equivalent (MET) hours per week (114.4, 95% CI=106.6 to 122.5) compared with lower class (140.3, 95% CI=132.3 to 148.5), lower middle class (167.4, 95% CI=163.8 to 171) and upper middle class (141.1. 95% CI 135.6 to 146.6).[48]

### Employment

The cross-sectional study conducted in a national representative sample in Malaysia revealed that employed individuals were more likely to participate in PA compared with unemployed individuals (likelihood ratio (LR)=0.035, p<0.05).[41] A global analysis of occupational structure and physical inactivity reported that individuals working in white-collar industry compared with agriculture were more likely to be physically inactive (OR=1.84, 95% CI=1.73 to 1.95).[52]

### Health insurance

In Malaysia, individuals who have insurance coverage were found to be 3.2% more likely to participate in PA, but do not spend more time than their counterparts who are not covered by insurance.[41]

### Attractiveness of leisure time PA/cost for leisure time PA

Attractiveness of recreational leisure time PA (LTPA) was more likely to predict the participation in LTPA among students of Tehran universities (OR=1.073), but time and money were not significant predictors.[50] Attractiveness of LTPA was specified as how participation in LTPA could promote diversity, excitement and competitiveness.

### Macroeconomic indicators

In Brazil, Gini index accounted for 12% of variance in PA (adjusted $R^2$=0.12); if Gini index of the city increases by 1%, the prevalence of sufficient PA would decrease by 41.5% for girls and 60.2% for boys.[45] Global analysis among 47

**Table 5** Evidence from economic studies related to PA in low/middle-income countries

| Evidence | Studies |
|---|---|
| **Economic burden** | |
| *Direct cost* | |
| UMC, 8886 (16.5%); LMC, 1366 (2.5%); LIC, 75 (0.1%) (INT$100 000, global %) in 2013 | Ding et al[38] |
| China, INT$3.075; Brazil, INT$1.634 billion in 2013 | Ding et al[38] |
| China, US$3.5 billion in 2007 | Zhang and Chaaban[43] |
| Brazil hospital admission cost, US$0.73 billion in 2013 | Bielemann et al[47] |
| Inpatient and outpatient cost of physical inactivity in China, US$0.4 billion and US$0.9 billion in 2000, respectively | Popkin et al[51] |
| *Indirect cost* | |
| UMC, 3814 (27.8%); LMC, 1350 (9.9%), 130 (0.9%) (INT$ 1 000 000, global%) in 2013 | Ding et al[38] |
| China, US$3.3 billion in 2007 | Zhang and Chaaban[43] |
| **Association of physical (in)activity and economic cost** | |
| *Association of physical (in)activity and direct health cost* | |
| Healthcare cost significantly higher for physically inactive individuals than active individuals | Turi et al,[40] Codogno et al,[42] Turi et al,[44] Patel et al[46] |
| Association of adherence to a PA programme by employees of a private healthcare company and direct medical costs is inconclusive | Mitsunaga et al[39] |
| *Association of physical (in)activity and productivity lost* | |
| Significant effect from PA on productivity loss-associated health problem | Araujo et al[36] |
| **Socioeconomic factors associated with PA participation and intensity** | |
| Negative association of income with PA | Cheah et al,[37 41] Abdi,[50] Sá et al,[49] Atkinson et al[52] |
| Employment status is positively associated with PA | Cheah et al[41] Chen et al[48] |
| Socioeconomic disparities are associated with PA | Sá et al[49] |
| Association of PA and macroeconomic indicators; <br>▶ PA is positively associated with HDI <br>▶ PA is negatively associated with Gini index | Atkinson et al,[52] Silva[45] |
| Attractiveness of LTPA is the predictor of participation of university students in PA over the time and cost factor of PA | Abdi[50] |
| **Economic evaluation studies** | |
| | Continued |

**Table 5** Continued

| Evidence | Studies |
|---|---|
| The cost–utility ratio was higher among those who reported taking medications (M) compared with the group with medication and regular PA: <br>▶ M only—US$3.92/QALY <br>▶ M+PA—US$3.21/QALY <br>▶ Now on medication or PA only—US$0.12/QALY <br>The incremental utility cost is advantageous for the medication+regular PA group | Queiroz et al[53] 2020 |

HDI, Human Development Index; LIC, low-income country; LMC, lower-middle-income country; LTPA, leisure time physical activity; PA, physical activity; QALY, quality-adjusted life year; UMC, upper-middle-income country.

LMICs reported that an increase in Human Development Index (HDI) value is associated with decreased levels of insufficient PA (OR=0.98, 95% CI=0.97 to 0.99).[52]

### Cost-effectiveness of PA interventions

A recent study conducted in Brazil revealed the value for money of combining PA and drug medication for the treatment of chronic diseases as the treatment scheme for >50-year-old attendees of selected primary healthcare settings.[53] Higher cost–utility ratio (US$3.92/QALY) was reported among individuals on 'medications only' compared with those on 'medication and regular PA' (US$3.21/QALY) and 'no medication or PA' groups (US$0.12/QALY).

### Research gaps

Box 1 summarises the gaps in the evidence base. The majority of the studies assessed either the socioeconomic factors associated with PA or the economic burden of physical inactivity. There is limited evidence on economic evaluation in LMICs.

### DISCUSSION

Economic burden of insufficient PA is consistently highlighted as a problem of high-income countries (HICs),

**Box 1  Gaps in evidence on economics of PA in LMICs**

▶ Economic evaluations of interventions and practices for PA promotion are minimal.
▶ A standard method to quantify economic burden of physical inactivity at country level is lacking.
▶ There is no standard method of operationalisation of physical inactivity to enable transferability of evidence.
▶ There is limited evidence available on economics of PA in LMICs compared with high-income countries.
▶ There is no evidence from locally led research on low-income countries.

LMICs, low/middle-income countries; PA, physical activity.

whereas a larger proportion of disease burden is a problem of LMICs.[38] Thus, the economic burden of physical inactivity in LMICs could be predicted to be escalated in the near future. However, considering LMICs as separate entity, even the current estimates of economic burden are huge in comparison with the income level of these countries.

The economic burden of insufficient PA[43 47] and the inverse association of PA with the economic cost of chronic diseases[36 40 42 44 46] were revealed by locally led research in this review. This is an important evidence to draw attention of authorities and policy-makers on PA promotion as a priority measure in LMICs. Global analysis on economic burden of insufficient PA including LMICs further enhanced this evidence.[38] However, locally led research was mostly contributed by Brazilian studies. The situation could be the same in other LMICs, although the evidence is limited from other countries. It is well evident that investing on PA promotion will pave the path towards achieving the sustainable development goals 2030 in LMICs.[1]

The cost estimates of studies from China and Brazil cannot be directly compared with the results from global analysis due to methodological differences and difference in year of cost estimate of these studies. There is no evidence on the economic burden of physical inactivity from LICs by local studies. Thus, capacity building of researchers of LMICs should be done to apply more consistent, robust methodology of cost estimation and to obtain better epidemiological and economic data for cost estimation, particularly in LMICs.[38]

The majority of research from economic evidence base in LMICs was focused on socioeconomic factors associated with participation in PA. These are also important to identify and prioritise population groups for PA promotion. Upper socioeconomic class, people in higher income category, unemployed individuals and white-collar job employees are at higher risk of less PA participation, low intensity and shorter duration of PA. Hence, these groups need to be prioritised in PA promotion in LMICs.

This review has shown that macroeconomic indicators of the country are significantly associated with PA of the population. PA level increases with the increase in HDI[52] and decreases in Gini index, which is a measure of inequality of income and wealth distribution of populations.[45] Although income level alone has shown a negative association with level of PA[37 41 48 49] in LMICs, the socioeconomic status of population subgroups as a whole has shown a positive association with the quality and quantity of PA in HICs.[54] This would have contributed to the positive association of level of PA and HDI of the countries.[45 52] Based on one study, we are unable to generalise the usefulness of health insurance schemes and attractiveness of PA options in PA promotion. However, more studies need to be conducted to conclude on the effect of health insurance on PA and effect of attractiveness of PA intervention compared with the time and cost variables.

It is of paramount importance to understand cost-effective interventions and strategies for PA promotion in LMICs. There was only one recent research on cost–utility analysis. This initiation to conduct economic evaluation studies in LMIC is an achievement. Prescribing 'PA along with medication' or 'PA alone' as a treatment regime for chronic diseases, which has proved to be advantageous through cost–utility analysis, should be considered to implement in LMICs. This is with great value for the treatment policy for NCDs in LMICs. The positive association of regular walking with less medical expenditure further supported the above evidence.[40] The health impact of exercise prescription was also well documented by studies from HICs.[55]

The economic evidence base on PA is biased towards middle-income countries (MICs), especially towards Brazil, as most of these research works were conducted in Brazil. Another global review on the economic burden of physical inactivity related to cardiovascular diseases also noted scarcity of research from LMICs other than few Brazilian studies.[17] Locally led research from LICs is not available. The same study population of >50-year-old adults seeking treatment from primary healthcare settings was studied in a majority of Brazilian studies. Adolescents as a study population need to be prioritised in the future research agenda on economic aspects of PA. This review showed that the economics of PA is a grossly neglected area in the research agenda of LMICs. Furthermore, locally led research in LMICs is much needed to overcome global health challenges.[56] Thus, more local research needs to be encouraged in this field in both MICs and LICs as a global priority to combat the global pandemic of physical inactivity. It is of paramount importance that researchers should explore and identify cost-effective interventions to improve PA within the context of LMICs. The health research capacity of LMICs is insufficient, and developing their capacity is not prioritised globally.[17] The overview of economic evaluations in LMICs reports the challenge of conducting economic evaluations in this context due to the absence of routine cost accounting system and limited patient information systems.[57] Furthermore, they reveal the limited capacity and funding to conduct health economic research in most LMICs.[57] An important suggestion that the findings of present review agree with the overview of economic evaluations in LMICs[57] is to encourage economic evaluation studies in LMICs by advocating funders to request economic evaluations to be included for trails and other evaluations to ensure funding for those research works.

Despite the limited number of studies in LMICs on economics of PA, they reported a good methodological quality. Although sensitivity analysis is not performed in the study on cost–utility analysis, it also reported a considerable quality, based on Drummond and Jefferson's checklist criteria.

The policy-makers and the authorities need to be advocated on the consistent evidence from LMICs on increased healthcare cost and productivity loss due to physical inactivity,[34 37 38 40 42 44] the possibility of escalating the economic burden related to physical inactivity in the near future[36] and the need to include PA promotion

across all policies. Furthermore, prioritisation and investment for research on economics of PA in LMICs need to be a global priority.

## Limitations

Meta-analysis of data was not possible in this review due to the wide variation in focus and methods of the studies included for the review. All locally led research was from MICs. Although we interpret the results for LMICs, there is a considerable socioeconomic disparity between LICs and MICs.

## CONCLUSIONS AND RECOMMENDATIONS

Considerable cost of chronic diseases and hospital admissions are connected to physical inactivity in LMICs. High income, high socioeconomic level, unemployment status and white-collar jobs are associated with low PA levels. PA should be prescribed alone or along with medication to reduce the cost per QALY among patients with chronic disease. Research on economics of PA is grossly inadequate in LMICs. It should be a priority research area in LMICs. Especially the research on economic evaluations related to PA should be considered as a priority in research agenda in LMICs.

**Contributors** NA and SP developed the idea for the review with inputs from PDR. PDR and NA conducted the review with the guidance of SP. PDR wrote the first draft. NA and SP revised the manuscript. NA will act as guarantor of the review.

**Funding** The authors have not declared a specific grant for this research from any funding agency in the public, commercial or not-for-profit sectors.

**Competing interests** None declared.

**Patient and public involvement statement** pateints or public not directly involved

**Patient consent for publication** Not required.

**Provenance and peer review** Not commissioned; externally peer reviewed.

**Data availability statement** All data relevant to the study are included in the article or uploaded as supplementary information.

**ORCID iD**
Priyanga Diloshini Ranasinghe http://orcid.org/0000-0001-8145-7288

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
