## [Reviewer comments · BMJ Open]

ARTICLE DETAILS

TITLE (PROVISIONAL)	The economics of physical activity in low and middle income countries: systematic review
AUTHORS	Ranasinghe, Priyanga; Pokhrel, Subhash; Anokye, Nana

VERSION 1 – REVIEW

REVIEWER	José Luis Márquez Andrade Universidad de Santiago de Chile, Chile.
REVIEW RETURNED	01-Mar-2020

GENERAL COMMENTS	The article by Dr. Ranasinghees and colleagues is interesting because it addresses a poorly known topic, the methodology is appropriate and rigorous and the most important findings are described. Some issues related to writing should be considered and are listed below: There is a lack of a link between physical activity and physical inactivity in the introduction, which makes it difficult to understand the real focus of the study, the introduction addresses physical inactivity and is fine, but all objectives are related to physical activity. Page 9 line 32. The number of articles does not match that registered in Figure 1 (17724). Page 10 line 13 to 18. The phrase is redundant, check the wording. Page 10, lines 33 to 35. "Four studies have estimated the economic burden of physical inactivity," the same sentence was written previously. Page 12 line 40. The subtitle "Evidence on economics and physical activity in LMIC" does not seem accurate because it indicates physical activity and the text is about physical inactivity. Page 15 line 6 to 11. I suggest modifying the wording to improve understanding of the paragraph. Page 15, lines 33 to 40. I suggest modifying the wording and using at least one comparative adjective to improve understanding. Page 15 line 60. The beta value should be checked and corrected. Page 16 line 3. The phrase "amount of participation decision" should be reviewed and corrected, the phrase is not clear and it would help to write it again. Page 16 lines 5 to 13. This phrase needs a major revision to show an inverse relationship between the occupation and the overall intensity of PA expressed and MET-hours / week. Page 16 line 13. The reference corresponds to appointment 46 not 48. Page 16 line 13 to 18. The inclusion of a number in parentheses in the middle of the sentence does not facilitate understanding, I suggest revising and avoiding the use of increment or decrease values using parentheses.
---

	Page 16, line 40 to 48. I suggest revising the wording and improving clarity in the expression of the main idea. Page 16 and 17 lines 60 to 6: The text should indicate "if the HDI of the city increased by 1%, the prevalence of physical activity would increase by 81.3% in boys", the sentence should be revised. Page 17 line 16 to 21. The phrase is redundant; it must be corrected.
--	---

REVIEWER	Melody Ding The University of Sydney, Australia
REVIEW RETURNED	08-May-2020

GENERAL COMMENTS	This systematic review summarised the literature on the economics of physical activity in low- and middle-income countries (LMICs). Overall, I think this systematic review (SR) addresses an important question and follows a relatively rigorous methodological protocol. However, the writing of the SR requires improvement. I have the following major comments. 1) Overall, the authors have conducted a comprehensive search and appraisal, and have systematically documented the process. However, the writing of the paper requires improvement. Throughout the paper, the definition and discussion regarding “economics” remains vague, the authors implied that one of the main motivations behind this SR is its policy implications. However, it is not clear to me what implications the paper would really have. 2) One of the reasons why I feel this way about this paper is that the Results and Discussion of the paper seemed to be mainly a collection of description of the studies found. As a scientific paper (particularly if the authors’ intention is to influence policy makers), the authors need to go the extra mile to extract higher-level information, identify patterns and synthesise results in a policy friendly ways, instead of just describing “2 studies used this design, 3 did this...”. 3) The authors still have not convinced me why a separate review is needed for LMICs, given that the overall literature has been reviewed by Ding et al. in 2017. The authors did use the word LMICs throughout (and mentioned the disease vs economic burdens distribution repeatedly), but I find the statement relatively generic. The authors should go deeper into questions such as: “why LMICs ?” “how do LMICs differ from HICs in terms of public health, policy making and the role economics play?” 4) On a related point, perhaps a more nuanced discussion regarding LMICs should also involve the heterogeneity within LMIC. The authors found studies mainly from UMICs, what is the implication here? 5) Another major issue: I am confused about the authors’ decision of lumping three types of studies together. The first type (economic burden) and second type (association between physical inactivity and economic costs), to me, are just different types of the same study –cost of illness analysis, or economic burden analysis. They used different methodological approaches (econometric vs PAF approach) and attempt to answer the same question. The third type is mainly exploring the socioeconomic correlates and predictors of physical activity, which to me, is a completely different research question. It is not intuitive to see them being lumped together without proper justification, and I even find the 3rd question distracting within the context of the current SR. ---- in fact, what is the authors’ definition of “economics” ?
--

	6) Related to Point 2)--- the Discussion is superficial and some of the conclusions do not seem to be based on the actual findings, e.g., the suggestion for Master degree training? Separate funding streams? These seem to be the authors' opinions rather than the interpretation of the findings. 7) The literature search is 2.5 years old. I highly recommend an update. 8) Overall, the quality of writing requires more attention. The paper was written in an unnecessarily lengthy way with quite a few points/sentences repeated throughout the paper. The readability and English language are limited at times. There are grammatical mistakes and unclear sentences throughout the paper. I suggest the authors consider editorial services during revision. Specific comments: 9) Abstract: the writing of the abstract should better follow the PRISMA statement in terms of "Provide a structured summary including, as applicable: background; objectives; data sources; study eligibility criteria, participants, and interventions; study appraisal and synthesis methods; results; limitations; conclusions and implications of key findings; systematic review registration number." ---key important information has been missed, for example, eligibility criteria, specific synthesis methods (description very generic). The description of results was very brief and the conclusions regarding disease burden vs economic burden were not the results from the current studies. 10) References in the Introduction are a bit outdated. For example, replace Ref 2 with Lee et al 2012 Lancet, or any of the most recent Global Burden of Disease reports. Physical activity prevalence references could be replaced with more updated data (Guthold 2018 Lancet Global Health for adults and Guthold 2019 Lancet Adolescent Health for children and adolescents) 11) The first sentence in the Introduction: do you mean "cause more deaths than..." 12) Eligibility criteria: intervention is not relevant for cost of illness studies 13) To me, research gaps should better fit the Discussion than Results. It is the authors' subjective interpretation of the study findings.
--	--

VERSION 1 – AUTHOR RESPONSE

Reviewers comment	How addressed	Location of revision
Reviewer 1		
1. There is a lack of a link between physical activity and physical inactivity in the introduction, which makes it difficult to understand the	We've rewritten the introduction as per the suggestions of both reviewers, establishing the link between physical activity and physical inactivity enabling better understanding to the reader on focus of the study.	Page 5-7 Definitions of PA & insufficient PA in page 5 last para

real focus of the study, the introduction addresses physical inactivity and is fine, but all objectives are related to physical activity.		
2. Page 9 line 32. The number of articles does not match that registered in Figure 1 (17724).	Systematic literature search retrieved 16724 records from the initial data base search, 11 records from other sources and 2962 records from updated search in 2020. Flow diagram of this process illustrated in figure 1. Figure 1 shows the number of articles by the source it was retrieved from. The cumulative number (16733) was given in the manuscript. Search was updated for 2020 as per the suggestion of reviewer2.	Page 12 para 1, figure 1 We have modified the text explaining the number of articles by source separately in par with the figure one to improve the clarity.
3. Page 10 line 13 to 18. The phrase is redundant, check the wording.	We omitted this sentence from the manuscript as the text above this sentence gives the same meaning.	Page 12 last paragraph
4. Page 10, lines 33 to 35. "Four studies have estimated the economic burden of physical inactivity," the same sentence was written previously.	We omitted the repeated sentence from the text.	Page 13 para 2
5. Page 12 line 40. The subtitle "Evidence on economics and physical activity	We have changed the subtopic as "Empirical results from reviewed studies" and text modified.	Page 15 para 1

in LMIC” does not seem accurate because it indicates physical activity and the text is about physical inactivity.		
6. Page 15 line 6 to 11. I suggest modifying the wording to improve understanding of the paragraph.	Two Malaysian studies revealed that individuals with higher levels of income, are less likely to participate in physical activity [37,41]. This was corroborated by a Brazilian study using data from Brazil’s National household sample survey 2008 to analyse income quintiles by active commuting. The study revealed that in urban population lowest income quintile had higher active transport (walking and cycling) level than the wealthiest population (2-5 times higher) [49] (p<0.001). Similarly, in China income levels were associated with lower physical activity [48].	We have modified the text Page 17 last para
7. Page 15, lines 33 to 40. I suggest modifying the wording and using at least one comparative adjective to improve understanding.	We have modified the text to improve meaning.	Page 18 para 1
8. Page 15 line 60. The beta value should be checked and corrected.	Modified the sentence.	Page 18, para 2
9. Page 16 line 3. The phrase "amount	Text modified. We have omitted the phrase “amount of participation decision”	Page 18, para 2

of participation decision" should be reviewed and corrected, the phrase is not clear and it would help to write it again.	which is not clear and included the likelihood of taking the decision to participate in PA into the manuscript to improve the clarity as suggested.	
10. Page 16 lines 5 to 13. This phrase needs a major revision to show an inverse relationship between the occupation and the overall intensity of PA expressed and MET-hours / week.	We omitted this sentence from the manuscript	Page 18
11. Page 16 line 13. The reference corresponds to appointment 46 not 48.	As the sentence is omitted this was removed.	Page 18
12. Page 16 line 13 to 18. The inclusion of a number in parentheses in the middle of the sentence does not facilitate understanding, I suggest revising and avoiding the use of increment or decrease values using parentheses.	A global analysis of occupational structure and physical inactivity reported that individuals working in white-collar industry compared to agriculture work were more likely to be physically inactive (OR=1.84, CI 1.73-1.95)[52] We have removed the parenthesis in the middle of the sentence and revised.	Page 18 para 2
13. Page 16, line 40 to 48. I suggest revising the wording and improving clarity in the expression	Attractiveness of recreational leisure time physical activity (LTPA) was more likely to predict the participation in LTPA among students of Teheran Universities (OR 1.073) but time and money	Page 19 para 1

of the main idea.	costs were not significant predictors[50]. We have revised the wording of the sentence in the text to improve the clarity.	
14. Page 16 and 17 lines 60 to 6: The text should indicate "if the HDI of the city increased by 1%, the prevalence of physical activity would increase by 81.3% in boys", the sentence should be revised.	In Brazil, Gini index accounted for 12% variance in Physical activity (adjusted R2=0.12), if Gini index of the city increase by 1%, prevalence of sufficient physical activity would decrease by 41.5% for girls and 60.2% for boys [45]. Global analysis among forty-seven LMIC, reported that increase of Human Development Index (HDI) value is associated with decreased levels of insufficient physical activity (OR=0.98, 95%CI 0.97-0.99) [52]. We have revised the sentence in the text.	Page 19 para 2
15. Page 17 line 16 to 21. The phrase is redundant; it must be corrected.	Box 1 summarises the gaps in the evidence base. Majority of the studies either assessed the socioeconomic factors associated with physical activity or the economic burden of physical inactivity. Limited evidence on economic evaluation in LMIC. We have modified the text on Research gaps to improve the clarity.	Page 20 para 1
Reviewer 2		
1) Overall, the authors have conducted a comprehensive search and	We have used broad criteria to consolidate all research conducted in LMIC as they are limited and	Page 6 introduction Page 21-25 (Discussion modified)

appraisal, and have systematically documented the process. However, the writing of the paper requires improvement. Throughout the paper, the definition and discussion regarding “economics” remains vague, the authors implied that one of the main motivations behind this SR is its policy implications. However, it is not clear to me what implications the paper would really have	scattered. (justified in introduction) Though, one of the objectives of this review is to identify potential target variables and cost-effective interventions for policy implications, locally led research on this regard were limited. However, Identification of potential target variables for policy and the gaps in research on economic of physical activity was gleaned from descriptive analysis of available evidence.	
2) One of the reasons why I feel this way about this paper is that the Results and Discussion of the paper seemed to be mainly a collection of description of the studies found. As a scientific paper (particularly if the authors’ intention is to influence policy makers), the authors need to go the extra mile to extract higher-level information, identify patterns and synthesise	We have modified and improved results and discussion as suggested. We have summarized the available evidence to fulfil the objective 1.	Results 12-20 Discussion 20-25

results in a policy friendly ways, instead of just describing “2 studies used this design, 3 did this...”.		
3) The authors still have not convinced me why a separate review is needed for LMICs, given that the overall literature has been reviewed by Ding et al. in 2017. The authors did use the word LMICs throughout (and mentioned the disease vs economic burdens distribution repeatedly), but I find the statement relatively generic. The authors should go deeper into questions such as: “why LMICs?” “how do LMICs differ from HICs in terms of public health, policy making and the role economics play?”	We have rewritten the introduction as per the suggestions of both reviewers and justified this fact too.	Page 6
4) On a related point, perhaps a more nuanced discussion regarding LMICs should also involve the heterogeneity within LMIC. The authors found	Economic evidence base on PA is biased towards MIC, especially towards Brazil as most of these researches were conducted in Brazil. Another global review on economic burden of physical inactivity related to cardiovascular diseases also noted scarcity of research	Discussion- page 23 Limitations-page 24

studies mainly from UMICs, what is the implication here?	from LMIC other than few Brazilian studies [56]. Locally led research from low income countries not available. Limitations: All locally led research were from MIC. Though we interpret the results for LMIC, there is a considerable socio-economic disparity between LIC and MIC.	
5) Another major issue: I am confused about the authors' decision of lumping three types of studies together. The first type (economic burden) and second type (association between physical inactivity and economic costs), to me, are just different types of the same study – cost of illness analysis, or economic burden analysis. They used different methodological approaches (econometric vs PAF approach) and attempt to answer the same question. The third type is mainly exploring the socioeconomic correlates and predictors of physical activity,	This current study has a broader scope in terms of the economic evidence (all types of economic analyses) but with limited geographic settings. In line with the standard definition of economics [12], we categorise economic research on physical activity in to; demand analyses of physical activity, impacts of physical (in) activity on economic outcomes such as costs, quality of life, labour market outcomes and economic evaluation studies on physical activity interventions [9,10]. This broad perspective is to facilitate a broader mapping of the multiple facets of the evidence base in LMIC. A specific review for LMIC is required considering the differing dynamics of the health systems [13] and the urgent need to better understand the research base and map gaps in knowledge given the growing burden of NCDs [6]. Health research in low- and middle-income countries (LMICs) is warranted to overcome global health challenges [14].	We have justified this fact in the introduction and the eligibility criteria of the methods section in the manuscript. (page 6-7 introduction) This rationale was explained in the protocol paper which was published in BMJ open [15] Ranasinghe PD, Pokhrel S, Anokye NK.(2019) The economics of physical activity in low-income and middle-income countries: protocol for a systematic review. BMJ Open, 9:e022686. doi: 10.1136/bmjopen-2018-022686

which to me, is a completely different research question. It is not intuitive to see them being lumped together without proper justification, and I even find the 3rd question distracting within the context of the current SR. ---- in fact, what is the authors' definition of "economics" ?		
6) Related to Point 2)--- the Discussion is superficial and some of the conclusions do not seem to be based on the actual findings, e.g., the suggestion for Master degree training? Separate funding streams? These seem to be the authors' opinions rather than the interpretation of the findings. 7) The literature search is 2.5 years old. I highly recommend an update. 8) Overall, the quality of writing requires more attention. The paper was written in an unnecessarily lengthy way with	We have rewritten the discussion as per suggestions of the reviewers. Literature search was updated for 2020 All three authors revised the manuscript to improve the quality of writing. Grammatical mistakes and unclear sentences were corrected as per suggestions.	Page 20-25 Figure 1, page 8 para 2

quite a few points/sentences repeated throughout the paper. The readability and English language are limited at times. There are grammatical mistakes and unclear sentences throughout the paper. I suggest the authors consider editorial services during revision.		
9) Abstract: the writing of the abstract should better follow the PRISMA statement in terms of "Provide a structured summary including, as applicable: background; objectives; data sources; study eligibility criteria, participants, and interventions; study appraisal and synthesis methods; results; limitations; conclusions and implications of key findings; systematic review registration number." ---key important information has been missed, for example, eligibility criteria,	We have rewritten the abstract structured according to PRISMA statement	Page 2-3

specific synthesis methods (description very generic). The description of results was very brief and the conclusions regarding disease burden vs economic burden were not the results from the current studies		
10) References in the Introduction are a bit outdated. For example, replace Ref 2 with Lee et al 2012 Lancet, or any of the most recent Global Burden of Disease reports. Physical activity prevalence references could be replaced with more updated data (Guthold 2018 Lancet Global Health for adults and Guthold 2019 Lancet Adolescent Health for children and adolescents)	We have updated the search as suggested and included latest articles for the review [figure 1]. Further, [4],[7],[8] references added as suggested.	List of references Page 26
11) The first sentence in the Introduction: do you mean “cause more deaths than...”	We have omitted this sentence and introduction revised.	
12) Eligibility criteria: intervention is not relevant for cost	We have included cost of illness studies, economic evaluation studies, intervention studies,	Page 9

of illness studies	descriptive studies - economic correlates of PA. One main objective of this review is to summarize available evidence on economic aspects of PA in LMIC	
13) To me, research gaps should better fit the Discussion than Results. It is the authors' subjective interpretation of the study findings.	Identifying research gaps and providing recommendations for research agenda in LMIC is one of the objectives of the study. Therefore, research gaps briefly summarized in the results section and further discussed under the "discussion" section.	Page 9 Para 1

VERSION 2 – REVIEW

REVIEWER	José Luis Márquez Universidad de Santiago de Chile, Chile.
REVIEW RETURNED	14-Sep-2020

GENERAL COMMENTS	In this version of the article by Dr. Ranasinghe et al., the authors appropriately incorporated all the suggestions made in my previous review, which facilitates a better understanding of their work.
---

REVIEWER	Melody Ding The University of Sydney, Australia
REVIEW RETURNED	11-Sep-2020

GENERAL COMMENTS	The authors have taken the reviewers' suggestions on board and substantially revised the manuscript. I find the paper much improved, especially in terms of setting the stage in the Introduction and contextualising the findings in the Discussion. I commend the authors for the additional efforts put into the paper. A job well done. Still, a few minor comments for authors to consider: Introduction--  1) Introduction: I think it is more logical to define physical activity first, before discussing the prevalence, disease burden etc. 2) The authors cited Ref 2 to claim that "physical activity has the largest positive impact on the risk of all-cause mortality" I think this conclusion may be taken out of the context and may provoke controversy. Suggest changing to "one of the largest..." 3) First paragraph "NCD's causes deaths than all other causes..." I do not understand this sentence. I think there may be a critical typo 4) Please note that WHO GAPP target has been revised to a relative reduction of 15% by 2030
--

	5) Second last paragraph of the Introduction: what is “demand analysis”? This term has not been mentioned elsewhere in the paper. Could you please define/clarify? Methods-- 6) Methods: “eligibility criteria and screening” note that “measures of association of any economic variable with PA” is strictly not an “outcome” Results--- 7) “Methods of reviewed studies”: “physical activeness” sounds a bit unconventional, suggest replacing with “physical activity” or “sufficient physical activity” 8) “Economic factors associated with physical (in) activity in LMIC”: note that income is also a measure of socioeconomic status, so it is confusing for me to have two separate sections on “income” and “socioeconomic status” ---plus, the authors need to define what “upper socioeconomic status” means in the Chinese study 9) In the same section, when discussing socioeconomic correlates with PA, particularly within the LMIC context, please clearly define the domains of physical activity. It has been well documented that the relationship between socioeconomic status and PA is domain-specific in LMIC. 10) What do the authors mean by “attractiveness of recreational leisure time PA”?
--	--

VERSION 2 – AUTHOR RESPONSE

Comments	How addressed?	Location of revision
-Please revise the ‘Strengths and limitations’ section of your manuscript (after the abstract). This section should contain five short bullet points, no longer than one sentence each, that relate specifically to the methods. The results of the study should not be summarised here.	Revised accordingly	Page 4
1) Introduction: I think it is more logical to define physical activity first, before discussing the prevalence, disease burden etc.	Revised accordingly	Page 5

2) The authors cited Ref 2 to claim that “physical activity has the largest positive impact on the risk of all-cause mortality” I think this conclusion may be taken out of the context and may provoke controversy. Suggest changing to “one of the largest...”	Revised accordingly	Page 5
3) First paragraph “NCD’s causes deaths than all other causes...” I do not understand this sentence. I think there may be a critical typo	Sentence revised accordingly	Page 5
4) Please note that WHO GAPPA target has been revised to a relative reduction of 15% by 2030	Statistic and date changed	Page 6
5) Second last paragraph of the Introduction: what is “demand analysis”? This term has not been mentioned elsewhere in the paper. Could you please define/clarify?	Demand analyses has been replaced with ‘examining socio-economic determinants of physical activity’	Page 7
6) Methods: “eligibility criteria and screening” note that “measures of association of any economic variable with PA” is strictly not an “outcome” Results---	Deleted as appropriate	Page 9
7) “Methods of reviewed studies”: “physical activeness” sounds a bit unconventional, suggest replacing with “physical	Physical activeness is replaced with ‘sufficient physical activity’.	Page 13

activity” or “sufficient physical activity”		
8) “Economic factors associated with physical (in) activity in LMIC”: note that income is also a measure of socioeconomic status, so it is confusing for me to have two separate sections on “income” and “socioeconomic status” --- plus, the authors need to define what “upper socioeconomic status” means in the Chinese study	The following revisions were done: The sections on income and socioeconomic status were combined into one section- socio economic status: upper socioeconomic status was defined as ≥ 12 index scores – score was based on educational attainment, occupation and income per capita	Page 16/17
9) In the same section, when discussing socioeconomic correlates with PA, particularly within the LMIC context, please clearly define the domains of physical activity. It has been well documented that the relationship between socioeconomic status and PA is domain-specific in LMIC.	Active transport was the domain specification provided in the reviewed paper. In addition, they looked at overall intensity of physical activity in MET hours per week	Page 17
10) What do the authors mean by “attractiveness of recreational leisure time PA”?	Definition of a Attractiveness of LTPA was provided as: Attractiveness of LTPA was specified in the reviewed paper as how participation in LTPA could promote diversity, excitement and competitiveness.	Page 18